# Seroprevalence and epidemiology of hepatitis B and C viruses in pregnant women in Spain. Risk factors for vertical transmission

**Ángeles Ruiz-Extremera**[1,2,3,4], **María del Mar Díaz-Alcázar**[1,3]*, **José Antonio Muñoz-Gámez**[1], **Marta Cabrera-Lafuente**[5], **Estefanía Martín**[6], **Rosa Patricia Arias-Llorente**[7], **Pilar Carretero**[1], **José Luis Gallo-Vallejo**[6], **Francisca Romero-Narbona**[8], **M. A. Salmerón-Ruiz**[5], **Clara Alonso-Diaz**[9,10], **Rafael Maese-Heredia**[8], **Lucas Cerrillos**[11], **Ana María Fernández-Alonso**[12], **Carmen Camarena**[5], **Josefa Aguayo**[11], **Miguel Sánchez-Forte**[12], **Manuel Rodríguez-Maresca**[12], **Alfredo Pérez-Rivilla**[9], **Rosa Quiles-Pérez**[1], **Paloma Muñoz de Rueda**[1,2,4], **Manuela Expósito-Ruiz**[4,13,14], **Federico García**[1,4,15], **Fernando García**[1,4,15], **Javier Salmerón**[1,2,3,4]

1 Hospital Universitario San Cecilio, Granada, Spain, 2 CIBER de Enfermedades Hepáticas y Digestivas (CIBEREHD), CIBER, Madrid, Spain, 3 Universidad de Granada, Granada, Spain, 4 Instituto de Investigación Biosanitaria Ibs.GRANADA, Granada, Spain, 5 Hospital Universitario La Paz, Madrid, Spain, 6 Hospital Universitario Virgen de las Nieves, Granada, Spain, 7 Hospital Universitario Central de Asturias, Oviedo, Spain, 8 Hospital Universitario Virgen de la Victoria, Málaga, Spain, 9 Hospital Universitario Doce de Octubre, Madrid, Spain, 10 RED SAMID (ISCIII ref. RD/16/0022), Spain, 11 Hospital Universitario Virgen del Rocío, Sevilla, Spain, 12 Hospital Universitario Torrecárdenas, Almería, Spain, 13 Unidad de Apoyo a la Investigación, Hospital Universitario Virgen de las Nieves, Granada, Spain, 14 Fundación para la Investigación Biosanitaria de Andalucía Oriental (FIBAO), Granada, Spain, 15 RED de SIDA (ISCIII ref. RD/16/0025/0040), Spain

* mmardiazalcazar@gmail.com

**Data Availability Statement:** All relevant data are within the paper.

## Abstract

### Background & aim

Worldwide, measures are being implemented to eradicate hepatitis B (HBV) and C (HCV) viruses, which can be transmitted from the mother during childbirth. This study aims to determine the prevalence of HBV and HCV in pregnant women in Spain, focusing on country of origin, epidemiological factors and risk of vertical transmission (VT).

### Methodology

Multicentre open-cohort study performed during 2015. HBV prevalence was determined in 21870 pregnant women and HCV prevalence in 7659 pregnant women. Epidemiological and risk factors for VT were analysed in positive women and differences between HBV and HCV cases were studied.

### Results

HBV prevalence was 0.42% (91/21870) and HCV prevalence was 0.26% (20/7659). Of the women with HBV, 65.7% (44/67) were migrants. The HBV transmission route to the mother was unknown in 40.3% of cases (27/67) and VT in 31.3% (21/67). Among risk factors for VT, 67.7% (42/62) of the women had viraemia and 14.5% (9/62) tested HBeAg-positive. All

**Funding:** This study received financial assistance from the following: Ciberehd, Fondo de Investigaciones Sanitarias del Instituto de Salud Carlos III. ISCIII, Proyecto del Plan Nacional I+D+i 2013-2016 (PI13/01925), Confinanciación Fondos FEDER. Gilead Fellowship Program (GLD14-00292 and GLD15-00307).

**Competing interests:** The authors have declared that no competing interests exist.

of the neonates born to HBV-positive mothers received immunoprophylaxis, and none contracted infection by VT. In 80% (16/20) of the women with HCV, the transmission route was parenteral, and nine were intravenous drug users. Viraemia was present in 40% (8/20) of the women and 10% (2/20) were HIV-coinfected. No children were infected. Women with HCV were less likely than women with HBV to breastfeed their child (65% vs. 86%).

## Conclusions

The prevalences obtained in our study of pregnant women are lower than those previously documented for the general population. Among the women with HBV, the majority were migrants and had a maternal family history of infection, while among those with HCV, the most common factor was intravenous drug use. Despite the risk factors observed for VT, none of the children were infected. Proper immunoprophylaxis is essential to prevent VT in children born to HBV-positive women.

## Introduction

According to the World Health Organization (WHO) [1], chronic infection by hepatitis B virus (HBV) affects 257 million people. It is particularly prevalent in sub-Saharan Africa and in East Asia, affecting 5–10% of all adults, while in North America, less than 1% are infected. The estimated prevalence is less than 5% in eastern Europe, 1.5% in northern Europe, 2% in southern Europe and 1% in western Europe. However, these values are probably underestimates because, among other reasons, the data are obtained from blood donors [2]. Moreover, the situation in Europe may be adversely affected by changing patterns of migration [3,4]. According to a systematic review conducted by the European Centre for Disease Prevention and Control (ECDC) [5] based on articles published from 2005 to 2015, the prevalence in Spain was around 0.8% (0.6–1.1) and according to the latest report available [3] it was 0.66% (0.34–0.97).

The WHO has reported that 71 million people are infected with hepatitis C virus (HCV) [1] and that the most severely affected regions are Europe and the Eastern Mediterranean (2.3%). In Europe, the prevalence of HCV antibodies (anti-HCV) is estimated at 1.1% (95%CI 0.9–1.4) [5]. In Spain, the prevalence of antibodies is 1.7% and that of viraemia (HCV RNA positive) is 1.2% [6]. In the Ethon Cohort [7], the prevalence of anti-HCV positive (anti-HCV+ve) was 1.23% and only 0.32% had viraemia. However, this prevalence might be affected by changes in patterns of migration in Europe, especially in Spain [8].

In Spain, the prevalence of HBV infection (surface antigen positive) in pregnant women ranges from 0.1% to 4.4% [5,9,10]. The vertical transmission (VT) of HBV continues to be one of the main routes of infection worldwide [11], especially in areas where it is endemic [12]. High viral load (VL), which mainly affects HBeAg-positive (HBeAg+ve) women, is the most important risk factor, although women who are HBeAg-negative (HBeAg-ve) and who present high VL are also at risk of transmission [13]. In this population, 90% chronicity of infected newborns has been reported [14].

The prevalence of anti-HCV in pregnant women in Spain is estimated to be 0.5–1.4% [9,15,16], and the prevalence of viraemia 42–72%. The rate of VT among women with HCV is low, 1–8% in non-coinfected mothers and around 20% in those coinfected with HIV. However, 90% of infected children acquire the virus by VT. The factors related to the VT of HCV

are VL and HIV coinfection [17–19]. No other factors related to childbirth, epidemiology or breastfeeding have been associated with VT.

Screening for HBV in pregnancy is routinely recommended, as immunoprophylaxis against VT is feasible [12,20]. According to 2015 data, generalised immunoprophylaxis at birth has reduced the proportion of chronically-infected children aged under five years from 4.7% to 1.3% [1]. Nevertheless, up to 10% of newborns whose mothers present high VL become infected [14,21]. Accordingly, it is recommended that in HBV+ve women, VL should be assessed at 24 weeks of pregnancy and antiviral treatment administered if the VL is high. On the other hand, the determination of HCV in pregnancy is only recommended for at-risk populations such as intravenous drug users, women with high-risk sexual practices, women with HIV coinfection and women born in countries with high endemicity for HCV. In consequence, the prevalence reported in the literature may not correspond to reality. Prophylactic and therapeutic measures to eradicate HBV and HCV are currently being implemented. Children infected by VT should be included in this programme, but screening during pregnancy for HCV is not yet universal, and therefore some children who acquire the virus through VT will remain undiagnosed.

In view of these considerations, the main aim of this study is to determine the prevalence of HBV and HCV in pregnant women in Spain, taking into account their country of origin, epidemiological factors and the risk of VT.

## Material and method

### Patients

To determine the prevalence of HBV and HCV among a population of pregnant women, a multicentre prospective open-label cohort study was carried out from January to December 2015. The following hospitals took part: Hospital Universitario 12 de Octubre Madrid (HU12O), Hospital Universitario La Paz Madrid (HULP), Hospital Universitario San Cecilio Granada (HUSC), Hospital Universitario Virgen de las Nieves Granada (HUVN), Hospital Universitario Central de Asturias Oviedo (HUCA), Hospital Universitario Torrecárdenas Almería (HUT), Hospital Universitario Virgen del Rocío Sevilla (HUVR) and Hospital Universitario Virgen de la Victoria Málaga (HUVV).

The population analysed in the HBV prevalence study consisted of 21870 pregnant women, whose data were provided by the researchers at each hospital, from an anonymised database managed by its Microbiology Service, which compiles and manages these data as usual clinical practice in pregnancy. After the delivery, the mothers found to be HBV-positive (HBV+ve) were invited to participate in the study, by completing the epidemiological survey, stating their country of origin and providing details of the delivery. In every case, the mother's informed consent was obtained. In addition, the donation of blood samples from each mother and child was requested, together with consent for medical follow-up of the neonate for 18 months. Fig 1 shows the flow chart for the recruitment of these participants.

In Spain, the determination of HCV does not form part of usual clinical practice, and so informed consent in this respect was requested and obtained in every case included in this study. Accordingly, tests for HCV were performed on 7659 women in week 12 of pregnancy, during the scheduled hospital visit for the detection of chromosomopathies. After the delivery, the mothers found to be HCV positive (HCV+ve) were invited to participate in the study, by completing the epidemiological survey, stating their country of origin and providing details of the delivery. In every case, informed consent in this respect was obtained. In addition, the donation of blood samples from the mother and child was requested, together with consent for

| Hepatitis B virus prevalence study: 21,870 pregnant women | | Hepatitis C virus prevalence study: 7,659 pregnant women |
|---|---|---|

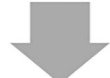

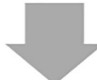

| 67 HBV-positive women agreed to complete a questionnaire on epidemiologic data, country of origin and characteristics of the delivery | | 20 HCV-positive women agreed to complete a questionnaire on epidemiologic data, country of origin and characteristics of the delivery |
|---|---|---|

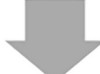

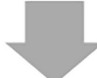

| 62 women provided blood samples | | The 20 women provided blood samples |
|---|---|---|

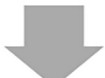

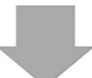

| For 54 of the 69 neonates (including two twin births), consent was sought to obtain samples from the children and to conduct follow up for 18 months. | | For 11 of the 20 neonates, consent was sought to obtain samples from the children and to conduct follow up for 18 months. |
|---|---|---|

**Fig 1. Flow chart for selection of participants.** This figure shows the flow chart of pregnant women recruitment for prevalence, epidemiological and risk factors for vertical transmission studies.

medical follow-up of the neonate for 18 months. Fig 1 shows the flow chart for the recruitment of these participants.

The inclusion criteria were HBV surface antigen positive (HBsAg+ve) or anti-HCV+ve status confirmed by enzyme-linked immunosorbent assay (ELISA) and the provision of informed consent.

## Epidemiological study

The study variables were participant's age, country of origin and risk factors for infection. The risk factors considered were transfusion, intravenous drug abuse (IVDA), tattoos, piercing,

surgery, dental treatment, infected partner, high-risk sexual practices, history of infection in the participant's mother or siblings and HIV coinfection, categorised as yes, no or unknown. The survey also asked whether the woman knew about the infection prior to her pregnancy. The route of transmission of infection was categorised as vertical, parenteral, sexual or unknown.

### Risk factors for vertical transmission

Antigen E (HBeAg) and antibody E (anti-HBe) were analysed for HBV, and VL was analysed for HBV and/or HCV.

The HBV+ve women were classified according to the EASL Clinical Practice Guidelines [22] as HBeAg-positive chronic infection (HBeAg+ve and DNA $>10^7$ IU/mL), HBeAg-positive chronic hepatitis (HBeAg+ve and DNA $10^4$–$10^7$ IU/mL), HBeAg-negative chronic infection (HBeAg-ve and DNA <2000 IU/mL) and HBeAg-negative chronic hepatitis (HBeAg-ve and DNA >2000 IU/mL).

Other information compiled for the study included data on the pregnancy and delivery, including in vitro fertilisation, antiviral treatment for HBV during pregnancy, type of delivery, gestational age, time elapsed since the breaking of the waters, weight, Apgar score of the neonate at birth and the type of lactation.

The study protocol is the same as has been described in previous studies by this research group (17–19).

### Analysis of the differences between HBV and HCV

The dependent variable was HBV+ve versus HCV+ve. The remaining variables were assumed to be independent.

### Virologic assays

The serological determination of HBsAg, anti-HCV and anti-HIV was carried out by commercial ELISA, which is routinely used in the laboratories of each of the participating hospitals.

The VL of HBV and/or HCV was determined by COBAS TaqMan (cut-off <12 IU/mL, <15 IU/mL. Roche Diagnostics, respectively), distinguishing between not detected, below cut-off but not quantifiable and greater than cut-off and quantifiable. In the HBV+ve women, HBeAg and anti-HBe were also determined.

### Statistical analysis

In our statistical analysis, the quantitative variables are described by the mean and the standard deviation, or in cases of non-normal distribution, by the median and the interquartile range. The qualitative variables are presented as absolute and relative frequencies. The corresponding prevalences and 95% confidence intervals were also calculated. The inter-group differences were determined by bivariate analysis, using Pearson's chi-square test or Fisher's exact test for the qualitative variables and Student's t test or the Mann-Whitney U test for the continuous ones. The normality of the data distribution was examined by the Kolmogorov-Smirnov test. A p-value <0.05 was considered significant. All data analyses were performed using IBM SPSS 19 statistical software.

### Ethical considerations

This study was carried out in accordance with the ethical guidelines of the Declaration of Helsinki of 1975, revised in 2013. All participants gave written and verbal informed consent. The

**Table 1. Seroprevalence of HBV and HCV among pregnant women at each hospital.**

| HBV | Positive cases | Women (n) | Prevalence (%) | 95%CI |
|---|---|---|---|---|
| HU12O[a] | 18 | 4224 | 0.42 | 0.22–0.62 |
| HULP[b] | 17 | 2901 | 0.59 | 0.31–0.86 |
| HUSC[c] –HUVN[d] | 9 | 2262 | 0.39 | 0.14–0.66 |
| HUCA[e] | 1 | 2362 | 0.04 | 0.01–0.12 |
| HUT[f] | 13 | 2037 | 0.64 | 0.29–0.98 |
| HUVR[g] | 29 | 5903 | 0.49 | 0.32–0.67 |
| HUVV[h] | 4 | 2181 | 0.18 | 0.004–0.362 |
| **Total** | **91** | **21870** | **0.42** | **0.33–0.50** |
| HCV | Positive cases | Women (n) | Prevalence (%) | 95%CI |
| HULP[b] | 7 | 2901 | 0.24 | 0.06–0.42 |
| HUSC[c] –HUVN[d] | 2 | 903 | 0.22 | 0.01–0.53 |
| HUCA[e] | 5 | 1713 | 0.29 | 0.04–0.55 |
| HUT[f] | 3 | 1671 | 0.18 | 0.01–0.38 |
| HUVV[h] | 3 | 471 | 0.64 | 0.01–1.34 |
| **Total** | **20** | **7659** | **0.26** | **0.15–0.38** |

[a]HU12O: Hospital Universitario 12 de Octubre, Madrid.

[b]HULP: Hospital Universitario La Paz, Madrid.

[c]HUSC: Hospital Universitario San Cecilio, Granada.

[d]HUVN: Hospital Universitario Virgen de las Nieves, Granada.

[e]HUCA: Hospital Universitario Central de Asturias, Oviedo.

[f]HUT: Hospital Universitario Torrecárdenas, Almería.

[g]HUVR: Hospital Universitario Virgen del Rocío, Sevilla.

[h]HUVV: Hospital Universitario Virgen de la Victoria, Málaga.

study protocol was firstly approved by Comité Ético de Investigación Provincial de Granada (Ethics Committee of the Principal Investigator's hospital) and later by the Ethics Committee at each participating hospital. The relevant provisions of Spanish data protection legislation were respected in all phases of the study.

## Results

### Prevalence of HBV and HCV in pregnancy

The prospective cohort for the HBV study consisted of 21870 pregnant women, of whom 91 were HBsAg+ve, with a prevalence of 0.42% (95% CI 0.33–0.50). Table 1 shows the distribution among the participating hospitals. The cohort for the HCV study contained 7659 women, of whom 20 were anti-HCV+ve, with a prevalence of 0.26% (95% CI 0.15–0.38). The prevalence of HBV differed significantly among the hospitals (p = 0.013), lowest at HUCA and highest at HULP, HUT and HUVR. On the other hand, the inter-hospital differences for anti-HCV prevalence were not significant (p = 0.52).

### Epidemiology and risk factors for vertical transmission in HBV-positive women

Of the 67 women included in the study, 31 (46.3%) were aware of the infection prior to pregnancy. None were coinfected with HCV, and there was one case (1.5%) of coinfection with HIV. By country of birth, 22 (32.8%) were Spanish, followed in frequency by those from China and Eastern Europe (Fig 2). The risk factors for infection in these pregnant women are shown

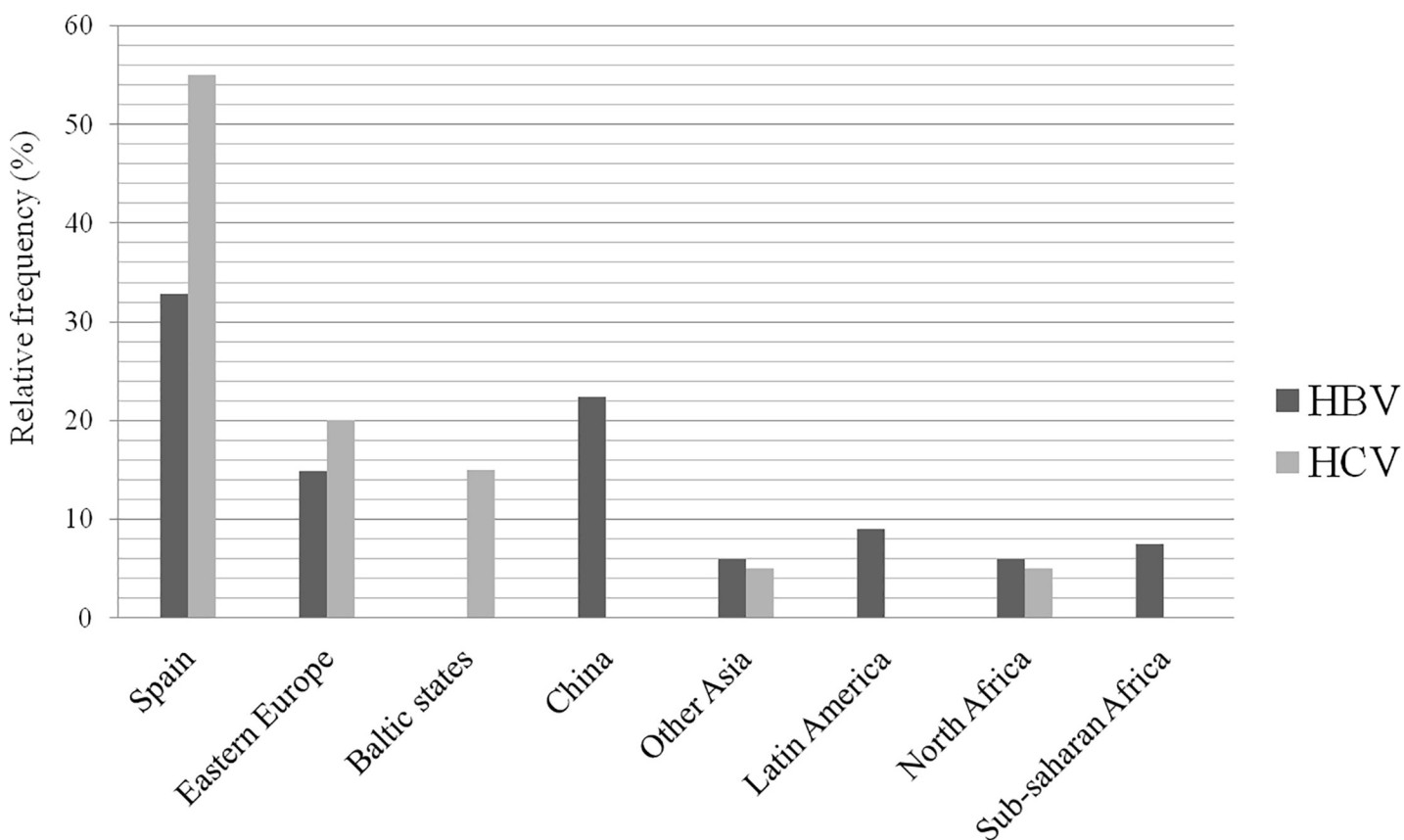

**Fig 2. Geographic origin of pregnant women with HBV or HCV infection.** This figure shows the geographic origin of the pregnant women with HBV or HCV infection who participated in the epidemiological study, expressed in relative frequency.

in Table 2. The route of transmission was parenteral in 14 cases (20.9%), vertical in 21 (31.3%), sexual in 3 (4.5%) and unknown in 27 (40.3%). Of the women presenting VT, 13 (61.9%) were of foreign origin.

Of the 62 HBsAg+ve women who provided samples, 42 (67.7%) had viraemia. Nine (14.5%) were HBeAg+ve and one (with a twin pregnancy) had chronic infection with a VL of 2070 IU/mL at the moment of delivery. Of the eight women who had HBeAg+ve chronic infection, six were treated during pregnancy: three achieved undetectable VL before delivery and in the other three the maximum VL was 1448 IU/mL. The two HBeAg+ve women who were not treated had a VL $>10^8$ IU/mL. Two of the HBeAg+ve women were Spanish, five were Chinese, one was from Central America and one was from sub-Saharan Africa. The ten neonates were monitored after birth, and none became infected. Of the 53 (85.5%) women who were HBeAg-ve, 40 had chronic infection and 13 had chronic hepatitis.

None of the pregnancies was achieved by in vitro fertilisation. The birth was spontaneous in 47 (72.3%) cases, by caesarean section in 13 (20%) (two of which were scheduled) and by instrumental delivery in five (7.7%). The data for the 69 neonates are summarised in Table 3. Only one of the low-weight children had a maternal history of antiviral treatment. Of the 30 neonates in whom the VL was analysed at birth, two were HBV+ve, although later tests were negative. All of the neonates born to HBsAg+ve women received immunoprophylaxis during the first twelve hours of life, for an average of 3.4 ± 3 hours. The ten children born to HBeAg

**Table 2. Epidemiology of pregnant women infected with HBV or HCV.**

|  | HBV n = 67 | HCV n = 20 | P |
|---|---|---|---|
| **Median age of the mother** [P25-P75] | 33 [28–37] | 34 [32.2–35.7] | 0.106 |
| **Aware of infection** | 31 (46.3%) | 13 (65%) | 0.128 |
| **Co-infection with HIV** | 1 (1.5%) | 2 (10%) | 0.125 |
| **Country of origin** |  |  |  |
| Spain | 22 (32.8%) | 11 (55%) | 0.070 |
| Other | 44 (65.7%) | 9 (45%) |  |
| **Transfusion** | 1 (1.5%) | 3 (15%) | 0.043 |
| **IVDA**[a] | 0 (0%) | 9 (45%) | <0.001 |
| **Tattoo** | 6 (9%) | 2 (10%) | 1 |
| **Piercing** | 3 (4.5%) | 1 (5%) | 1 |
| **Surgery** | 13 (19.4%) | 6 (30%) | 0.383 |
| **Dental treatment** | 4 (6%) | 1 (5%) | 1 |
| **Infected partner** | 3 (4.5%) | 3 (15%) | 0.143 |
| **High-risk sexual practices** | 3 (4.5%) | 3 (15%) | 0.148 |
| **Family history**[b] | 21 (31.3%) | 0 (0%) | 0.001 |
| **Unknown route of infection** | 27 (40.3%) | 4 (20%) | 0.066 |

[a]IVDA: Intravenous drug abuse.

[b]Family history: previous infection in the mother or siblings.

+ve women received immunoprophylaxis in the first three hours of life. None of the 54 children who completed the follow-up became infected.

## Epidemiology and risk factors for vertical transmission in HCV-positive women

Thirteen women (65%) were previously aware of their HCV infection. None were coinfected with HBV and there were two cases (10%) of coinfection with HIV. By country of origin, the majority of women were from Spain (11 cases, 55%) and Eastern Europe (four cases, 20%) (Fig 2). The risk factors for infection are detailed in Table 2. The route of transmission was parenteral in 16 women (80%), of whom nine had a history of IVDA. The route of transmission was unknown in four cases (20%).

HCV RNA was positive (HCV-RNA+ve) in eight (40%) of the 20 women, with an average VL of $1.9x10^5$ IU/mL ($4.4x10^5$–$3.5x10^6$ IU/mL). The distribution of genotypes was 1b (25%), 3 (25%), 4c (12.5%) and unknown (37.5%).

None of the pregnancies was achieved by in vitro fertilisation. Delivery was spontaneous in 12 cases (60%), by caesarean section in four (20%) and by instrumental delivery in four (20%). The data for the 20 neonates are summarised in Table 3. Five children of the eight women with viraemia completed the follow up and none became infected.

## Analysis of the differences between HBV and HCV-positive pregnant women

As can be seen in Table 2, 65.7% (44/67) of the women with HBV were foreign born, versus only 45% (9/20) of the women with HCV. Nevertheless, the difference was not statistically significant (p = 0.07). A similar pattern was observed for the unknown route of maternal infection, which was more frequent among the women with HBV than those with HCV (40.3% vs. 20%; p = 0.066). VT as the mechanism of infection was more frequent for the HBV+ve women

**Table 3. Characteristics of the neonates born to HBV or HCV-positive mothers.**

|  | HBV n = 69 | HCV n = 20 | P |
|---|---|---|---|
| Premature[a] | 8 (11.6%) | 4 (20%) | 0.265 |
| Weight (g) Mean ± standard deviation | 3194.7 ± 681 | 2998.8 ± 582 | 0.248 |
| Time elapsed since breaking of the waters (h) Mean ± standard deviation | 7 ± 8 | 12 ± 17 | 0.212 |
| Apgar score Median [P25-P75] | 9 [5–10] | 9 [6–10] | 0.834 |
| Breastfeeding | 55 (86%) | 13 (65%) | 0.044 |

[a]Premature: gestational age <37 weeks.

(31.3% vs. 0%; p = 0.001), while IVDA was more frequent in the HCV group (45% vs. 0%; p<0.001). Previous blood transfusion was also significantly higher in the HCV group (15% vs. 1.5%; p = 0.043).

There were no statistically significant differences among the neonates for any of the variables analysed, except that a significantly lower proportion of the women with HCV breastfed their children (65% vs. 86%; p = 0.044) (Table 3).

## Discussion

In our study population, the prevalence of HBV was 0.42% (91/21870) and that of HCV was 0.26% (20/7659). The HBV+ve women were more likely to be foreign born than those with HCV. The most frequent route of transmission of HBV was of unknown origin, while among those of known origin, VT was most commonly observed. Regarding the risk factors for VT, 67.7% of the women presenting HBV had viraemia and 14.5% were HBeAg+ve. Immunoprophylaxis was administered correctly to all the neonates born to HBV+ve mothers. Among the HCV+ve mothers, the parenteral route of transmission was most common, mainly due to IVDA. In the HCV group, 40% had viraemia and 10% were co-infected with HIV. The HCV +ve women were less likely than HBV+ve to breastfeed their children.

In this study, the observed prevalence of HBsAg+ve was significantly different among the participant hospitals. The highest level was obtained for the HUT in Almeria (0.64%), a hospital that serves a large migrant population from North and sub-Saharan Africa. In contrast, the data for anti-HCV prevalence were fairly consistent among the participant hospitals, with the sole exception of the HUVV in Malaga, where the prevalence was higher, although in this case the sample size was very small.

In the general population in Spain, the prevalence of HBV is 0.66% (0.34–0.97) [3], which is higher than the prevalence we found in pregnant women. In a previous study conducted in 1986–89 [23] at the HUSC in Granada, based on a population of 4450 pregnant women, the prevalence of HBV was 1.53% (95% CI 1.14–1.92). Thirty years later, according to our findings, the prevalence in this city has fallen to 0.39%. For comparison, a 2009 study by Salleras et al. [24], with a population of pregnant women in Catalonia, and which has been taken as a benchmark for Spain by the ECDC [5], recorded a very low prevalence, of 0.1%.

Our findings show that 65.7% of the participants who were HBV+ve were foreign born, with a particularly strong presence of women from China and eastern Europe, which corroborates the pattern reported previously [25,26]. In a study conducted in Madrid [26], the highest rate of seropositivity was found among women from Romania and other Eastern European countries (23.7%), followed by those from China (20.3%) and sub-Saharan Africa (18.6%). In recent years, the prevalence of HBV in Spain has tended to decrease, and if the population considered excluded those of foreign origin, the prevalence would probably be even lower.

In most of the cases analysed in our study, the route of transmission of HBV infection was unknown, while the most common of the known routes was maternal family history of HBV infection (31.3%), although some women presented several risk factors for infection. A large majority of the women (67.7%) had viraemia and therefore were potential transmitters of infection. 14.5% were HBeAg+ve, which increased the risk of transmission still further. However, only six of these women received antiviral treatment during their pregnancy, which shows that not all hospitals follow clinical practice guidelines in this respect [22]. A previous study carried out in Granada with 4169 pregnant women [25] reported that HBV was present in 27, of whom only one (a woman of Asian origin) was HBeAg+ve.

All of the 69 neonates born to HBV+ve mothers received immunoprophylaxis during the first twelve hours of life, and the ten children born to HBeAg+ve women received immunoprophylaxis in the first three hours of life. None of the 54 neonates who completed the follow up became infected. This includes the two children of HBeAg+ve women with high VL who were untreated, which highlights the importance of correctly administering immunoprophylaxis to neonates at an early stage of life.

In our study, the prevalence of HCV was 0.26%, and only 40% of these cases were HCV-RNA+ve. On the other hand, previous studies in the general population have found a higher prevalence, of 1.7% (0.4–2.6), with HCV-RNA+ve in 68.6% of these cases [27]. Nevertheless, research has shown that the prevalence of HCV is lower among women than in the general population. In a study conducted in the United States [28], the prevalence in men was 1.56% versus 0.75% in women. In another study, also conducted in the USA [29], with 87924 pregnant women, the prevalence of HCV was 1.2%. In Navarre (Spain), a study of 7314 pre-surgical patients [30] revealed a prevalence of 0.94%, with higher values for men than for women (1.25% vs. 0.62%, p = 0.0049). Finally, a study carried out in 1993–95 with 3003 pregnant women at the HUSC in Granada [19] reported that only 19 were HCV+ve (0.63%) while 14 (74%) presented viraemia. Thus, the value of 0.26% obtained in the present study is much lower than these previous findings.

Among our study population, 55% of the women with HCV were Spanish, followed in frequency by those from Eastern Europe (20%), which is in line with previous findings [31]. According to earlier research [27], 80% of the cases recorded worldwide with positive viraemia are located in 31 countries, one of which is Spain, but over half of the infections correspond to just six countries: China, Pakistan, Nigeria, Egypt, India and Russia. Parenteral transmission is the most common route of infection, and most cases involve a history of IVDA, although in 20% of cases the origin of the infection is not known. This knowledge gap could justify the introduction of HCV screening in pregnancy. Similar findings have been reported in other studies [29,32,33]. Infection is significantly more frequent in people aged under 30 years, those who are of European descent, especially those from Eastern Europe, and those with a history of IVDA. The most frequent genotypes recorded in our study were 1b and 3, followed by 4, which is consistent with previous research findings [33]. Two women (10%) had co-infection by HIV, which increases the probability of HCV being transmitted to the child [17].

Among the children of women with HCV infection, 20% were premature, a value that is higher than in the general population. Moreover, only 65% initiated breastfeeding, which is well below the figure for children of HBV+ve mothers (86%) and the general population. These findings may be more influenced by the history of IVDA than by the HCV *per se* [34]. This may also be the reason why not all the children of mothers with viraemia completed the follow up.

Among the specific characteristics differentiating the HBV+ve women were the predominance of infection by VT and the larger proportion who breastfed their children. In addition,

more were foreign born, compared to the women with HCV. In the HCV group, the main distinguishing features observed were blood transfusion and a history of IVDA.

The main limitation of this study concerned the determination of HCV in the pregnant women, since screening for this condition is not universal in Spain during pregnancy, and therefore signed informed consent was required. For this reason, the sample size for our analysis of HCV prevalence was lower than that for HBV. Furthermore, some of the Spanish hospitals that were invited declined to participate. Another limitation concerned the follow-up of the children of seropositive mothers; many of these children were adopted after birth and so their place of residence changed.

In summary, the prevalences of HBV and HCV that we report are lower than those documented previously. A significant number of the women with HBV were foreign born and/or had a maternal family history of infection. Among those with HCV, many had a history of IVDA, which probably influenced the fact that these women were less likely to complete follow up and less likely to breastfeed their children. Over half of the women with HBV had viraemia, and therefore were potential transmitters of the infection to the neonate. Therefore, the provision of appropriate immunoprophylaxis (including immunoglobulin therapy and vaccination) is important in neonates born to HBV+ve mothers. However, among the women with HCV, despite the presence of viraemia and coinfection with HIV, there was no transmission to the children.

## Acknowledgments

We thank Pedro Lucas at FIBAO (Public Foundation for Biomedical Research in Eastern Andalusia) for designing the database used in this study.

This paper will be presented by Maria del Mar Diaz-Alcazar to obtain her Ph.D. within the doctoral programme 'Clinical Medicine and Public Health' at the University of Granada (Spain).

## Author Contributions

**Conceptualization:** Ángeles Ruiz-Extremera.

**Data curation:** María del Mar Díaz-Alcázar, José Antonio Muñoz-Gámez, Marta Cabrera-Lafuente, Estefanía Martín, Rosa Patricia Arias-Llorente, Pilar Carretero, José Luis Gallo-Vallejo, Francisca Romero-Narbona, M. A. Salmerón-Ruiz, Clara Alonso-Diaz, Rafael Maese-Heredia, Lucas Cerrillos, Ana María Fernández-Alonso, Carmen Camarena, Josefa Aguayo, Miguel Sánchez-Forte, Manuel Rodríguez-Maresca, Alfredo Pérez-Rivilla, Rosa Quiles-Pérez, Paloma Muñoz de Rueda, Federico García, Fernando García.

**Formal analysis:** Ángeles Ruiz-Extremera, Rosa Quiles-Pérez, Paloma Muñoz de Rueda, Manuela Expósito-Ruiz.

**Funding acquisition:** Ángeles Ruiz-Extremera, Javier Salmerón.

**Investigation:** Ángeles Ruiz-Extremera, María del Mar Díaz-Alcázar, Manuela Expósito-Ruiz.

**Methodology:** Ángeles Ruiz-Extremera, María del Mar Díaz-Alcázar, Manuela Expósito-Ruiz.

**Project administration:** Ángeles Ruiz-Extremera, José Antonio Muñoz-Gámez, Rosa Quiles-Pérez, Paloma Muñoz de Rueda, Javier Salmerón.

**Resources:** Ángeles Ruiz-Extremera.

**Software:** Ángeles Ruiz-Extremera, María del Mar Díaz-Alcázar, Manuela Expósito-Ruiz.

**Supervision:** Ángeles Ruiz-Extremera.

**Validation:** Ángeles Ruiz-Extremera.

**Visualization:** Ángeles Ruiz-Extremera.

**Writing – original draft:** Ángeles Ruiz-Extremera, María del Mar Díaz-Alcázar.

**Writing – review & editing:** Ángeles Ruiz-Extremera, María del Mar Díaz-Alcázar, Manuela Expósito-Ruiz, Federico García, Javier Salmerón.

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
