## [Decision Letter · Decision Letter 0]

7 Apr 2020

PONE-D-20-00132

Seroprevalence and epidemiology of hepatitis B and C viruses in pregnant women in Spain. Risk factors for vertical transmission.

PLOS ONE

Dear Mrs Díaz Alcázar,

Thank you for submitting your manuscript to PLOS ONE. After careful consideration, we feel that it has merit but does not fully meet PLOS ONE’s publication criteria as it currently stands. Therefore, we invite you to submit a revised version of the manuscript that addresses the points raised during the review process.

We would appreciate receiving your revised manuscript by May 22 2020 11:59PM. To enhance the reproducibility of your results, we recommend that if applicable you deposit your laboratory protocols in protocols.io, where a protocol can be assigned its own identifier (DOI) such that it can be cited independently in the future. For instructions see: http://journals.plos.org/plosone/s/submission-guidelines#loc-laboratory-protocols

We look forward to receiving your revised manuscript.

Kind regards,

Anna Kramvis

Academic Editor

PLOS ONE

Journal Requirements:

2. Please note that according to our submission guidelines (http://journals.plos.org/plosone/s/submission-guidelines), outmoded terms and potentially stigmatizing labels should be changed to more current, acceptable terminology. For example: “Caucasian” should be changed to “white” or “of [Western] European descent” (as appropriate).

3. Please include additional information regarding the survey or questionnaire used in the study and ensure that you have provided sufficient details that others could replicate the analyses. For instance, if you developed a questionnaire as part of this study and it is not under a copyright more restrictive than CC-BY, please include a copy, in both the original language and English, as Supporting Information. Moreover, please include more details on how the questionnaire was pre-tested, and whether it was validated.

Additional Editor Comments (if provided):

Please pay special attention to all the reviewers' comments and suggestions in particular paying special attention in improving the English language and grammar.

There are a number of errors even in the abstract: correct terminology for "E antigen" is HBeAg - please correct here and in the rest of the manuscript where it is referred to as "AgHBe". "VHB" should be HBV is assume, correct in the abstract and in all other sections of the manuscript. The comment of the prevalence being different to previous studies is not correct considering that here you only looked at prevalence in females. Please consider correcting this. The statement in the abstract "all neonates received immunoprophylaxis" I assume this is all neonates born to HBV+ve mothers! Thus please correct here and in the rest of the manuscript.

What is not clear is whether there is an overlap between the pregnant women tested for HBV and those for HCV. Please clarify!!

Where the ethical considerations the same for both HBV and HCV studies? Were separate protocols submitted to the institutional review boards?

Was the prevalence between the different hospitals significantly different? Could you explain why for example HUCA had 0.04% whereas HUT had 0.64% HBV prevalence? Were the population groups serviced different between the two hospitals? You do mention in your discussion that HUT had more pregnant women of foreign origin. Were they mainly from eastern Europe, China or sub-Saharan Africa? Another important analysis that would add value to your study, would be to compare the number of HBeAg-positivity between Spanish women, and the various immigrant populations, i.e. Chinese compared to European and African. It has important implications regarding the management of HBV infection. For example mother to child transmission is more frequent in Asian populations compared to Africans because of differences in the frequency of HBeAg-positivity in the two groups.

Reviewers' comments:

Reviewer's Responses to Questions

**Comments to the Author**

1. Is the manuscript technically sound, and do the data support the conclusions?

Reviewer #1: Yes

Reviewer #2: Yes

2. Has the statistical analysis been performed appropriately and rigorously? 

Reviewer #1: Yes

Reviewer #2: Yes

3. Have the authors made all data underlying the findings in their manuscript fully available?

Reviewer #1: Yes

Reviewer #2: Yes

4. Is the manuscript presented in an intelligible fashion and written in standard English?

Reviewer #1: Yes

Reviewer #2: No

5. Review Comments to the Author

Reviewer #1: The study by Ruiz-Extremera et al., entitled “Seroprevalence, epidemiology and clinical characteristics of hepatitis B and C viruses in pregnant women in Spain” aimed to study the prevalence of HBV/HCV in pregnant women, subsequent vertical transmission, and newborn immune prophylaxis etc. Additionally, the data was analyzed in view of human migration to identify other possible/associated risk factors.

The study is concise, addressing certain important issues like, the need for proper immune prophylaxis, probable nation wide vaccination program etc. Still further modifications are needed before deemed fit for acceptance. Some concerns that need to be addressed are as follows:

Some major points are:

1. The whole MS needs linguistic correction. For eg. In the abstract section: vide line no. 38-39 “HBV prevalence was analyzed in a population of 21,870 women and that of HCV, in one of 7,659 ”. In the line “that of HCV, in one of 7,659” does mean only one subject was positive out of 7659 subjects”.

I think a better way of representation is “HBV prevalence was analyzed in a population size of 21,870 women and for HCV, the study population size was 7,659 women”.

Another eg. Line no 83 , “chronification” should be “chronicity”.

2. For any data represented with percentages should accompany the numerical ration to ease things for understanding and more meaningful. Viz. line no. 43 “HBV prevalence was 0.42% and that of HCV, 0.26%” should be “HBV prevalence was 0.42% (91/21870) and that of HCV, 0.26% (20/7659)”. Also, it is better to mention the table number.

3. In the discussion section, vide line no. 298-299 “In the general population in Spain, the prevalence of HBV is 0.66% (0.34-0.97) (3), which is higher than was recorded in the present research”. As HBV chronicity display gander disparity, thus probably, this difference is expected. It will be better to comment on that.

4. This reviewer suggests including and discussing the importance of HBV vaccination in the conclusion section.

Reviewer #2: Line 38 Methodology. Multicentre open-cohort study performed during 2015. Which period of 2015?

The papers' english needs thorough grammar revision.

Examples ( Line 49. Viraemia was present in 40% and 10% were co-infected with HIV .

Line 75. only 0.32%. This prevalence could be modified by changes in patterns of migration

Line 76. towards the European Union in general and Spain, in particular (8).

Line 100. HIV coinfection or pregnant women from countries where HCV is endemic )

Figure 1 Diagram labelling is not very clear.

Figure 2 needs labelling

6. PLOS authors have the option to publish the peer review history of their article (what does this mean?). If published, this will include your full peer review and any attached files.

Reviewer #1: No

Reviewer #2: No

---

## [Author Response · Author response to Decision Letter 0]

3 May 2020

We would like to thank you for the suggestions provided regarding the article entitled “Seroprevalence and epidemiology of hepatitis B and C viruses in pregnant women in Spain. Risk factors for vertical transmission” by the authors Ángeles Ruiz-Extremera, María del Mar Díaz-Alcázar, José Antonio Muñoz-Gámez, et al. As requested, we provide more information in the rebuttal letter about the changes made to the manustript. 

We very much appreciate the comments and suggestions made, which have been very helpful in improving the manuscript. We hope these changes are considered appropriate and that the manuscript is now suitable for publication. In any case, we will gladly consider any further comments you may have.

---

## [Editor Report · Decision Letter 1]

7 May 2020

Seroprevalence and epidemiology of hepatitis B and C viruses in pregnant women in Spain. Risk factors for vertical transmission.

PONE-D-20-00132R1

Dear Dr. Díaz Alcázar,

We are pleased to inform you that your manuscript has been judged scientifically suitable for publication and will be formally accepted for publication once it complies with all outstanding technical requirements.

With kind regards,

Anna Kramvis

Academic Editor

PLOS ONE
---

## [Editor Report · Acceptance letter]

8 May 2020

PONE-D-20-00132R1 

Seroprevalence and epidemiology of hepatitis B and C viruses in pregnant women in Spain. Risk factors for vertical transmission. 

Dear Dr. Díaz-Alcázar:

I am pleased to inform you that your manuscript has been deemed suitable for publication in PLOS ONE. Congratulations! Your manuscript is now with our production department. 

With kind regards,

on behalf of

Prof. Anna Kramvis 

Academic Editor

PLOS ONE